# NeuralMatrix: Compute the Entire Neural Networks with Linear Matrix Operations for Efficient Inference

## Abstract

The inherent diversity of computation types within individual deep neural network (DNN) models necessitates a corresponding variety of computation units within hardware processors, leading to a significant constraint on computation efficiency during neural network execution. In this study, we introduce NeuralMatrix, a framework that transforms the computation of entire DNNs into linear matrix operations, effectively enabling their execution with one general-purpose matrix multiplication (GEMM) accelerator. By surmounting the constraints posed by the diverse computation types required by individual network models, this approach provides both generality, allowing a wide range of DNN models to be executed using a single GEMM accelerator and application-specific acceleration levels without extra special function units, which are validated through main stream DNNs and their variant models.

## 1 Introduction

In recent years, the development of various types of deep neural networks (DNNs) has found applications in a wide range of scenarios. As neural network architectures continue to expand in size and complexity, they pose substantial computational challenges, especially for resource-constrained platforms and budget-conscious organizations. Application-specific integrated circuits (ASICs) offer a promising solution for supporting DNNs on mobile and edge devices. For example, Bai et al. (2018) introduced a CNN accelerator design that incorporates a multiplier array, add tree, normalization, ReLU, and pooling units. Similarly, Thierry Tambe et al. (2021) proposed an edge transformer accelerator featuring processing units (with floating-point vector and accumulate) and dedicated function units for layer normalization, softmax, and other unique operators in each layer.

ASIC-based accelerators are known for their efficient execution of specific DNN applications. However, their inherent specificity, including the type and number of computation units, can restrict their adaptability from one DNN to another. For example, transformer-based BERT uses 72.5% of its computation cycles for versatile nonlinear operations (Thierry Tambe et al., 2021), necessitating the integration of specific types and amounts of nonlinear functional units in its accelerator. However, these functional units can become unnecessary burdens when the same accelerator is used for other networks, such as CNN and GNN, which have far fewer nonlinear operations. Consequently, a significant gap exists between the generality and computation efficiency of the accelerator when it runs versatile DNN applications (Geng et al., 2021).

In this study, we introduce NeuralMatrix, a framework that combines the best of both worlds, as illustrated in Fig. 1. On one hand, it overcomes the specificity limitations of computation types and enables the computation of versatile DNNs on a single general matrix multiplication (GEMM) accelerator. On the other hand, compared to other general-purpose processors such as CPUs and GPUs, NeuralMatrix achieves application-specific acceleration levels by converting DNNs to linear matrix operations and executing them with GEMM accelerators. Supporting different DNN architectures on a single GEMM accelerator is not trivial. Several challenges need to be addressed, including how to map different computation types in DNN computation to linear matrix operations so that a single GEMM accelerator can fully support them and how to eliminate the impact of linear matrix operations on DNN inference accuracy.

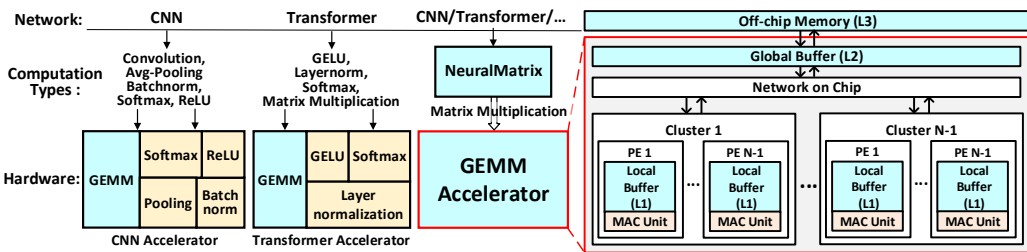

Figure 1: NeuralMatrix translates neural network computation tasks into matrix operations, enabling them to be fully supported by general matrix multiplication (GEMM) accelerators.

Our comprehensive experiments, featuring three popular categories (CNN, Transformers, and GNN) as illustrative backbone models, demonstrate that DNNs incur only an average of 1.32% accuracy loss when converted to linear matrix operations. Remarkably, this matrix-based neural network computation improves computation efficiency (i.e., throughput per power) by 49.58, 8.50, and 2.35 times when compared to the CPUs, GPU, and SoC, achieving levels of computing efficiency traditionally attainable only with carefully designed, application-specific accelerators tailored to one specific network model. To the best of our knowledge, we are pioneering the transformation of diverse entire DNNs into linear matrix operations, revealing the substantial advantages of generality and computational efficiency through the utilization of a single GEMM accelerator. Our innovative framework, NeuralMatrix, adeptly addresses the irregularities present in a wide range of neural network models, thus simultaneously achieving both generality and computational efficiency.

## 2 BACKGROUND AND RELATED WORK

### 2.1 INTENSIVE AND VERSATILE COMPUTATIONS IN DNNS

The computational demands of DNNs pose challenges for conventional platforms. Two types of computational platforms have been developed to accelerate DNNs. First, application-specific integrated circuits (ASICs) with dedicated functional units target specific network models for optimal efficiency (Xiaocong Lian et al., 2019; Tao Luo et al., 2017; Li et al., 2020; Khan et al., 2021; Wang et al., 2021). Second, versatile general-purpose processors like graphics processing units (GPUs) and tensor processing units (TPUs) accelerate DNNs at the cost of high power consumption due to numerous processing units (Wang et al., 2019).

Previous work has attempted to address versatility issues stemming from DNNs' nonlinear computations. Approaches such as piecewise linear (PWL) based approximations (Dong et al., 2020; Khan et al., 2021; Lyu et al., 2021) and neural network-based approximations (Yu et al., 2022) have been proposed to accelerate nonlinear operations in neural networks. An automated approximation framework (Lu et al., 2023) has been developed to simplify and automate this process, leveraging a neural network to approximate nonlinear operations. MA-BERT (Ming et al., 2022) replaces complex functions with computation-friendly ones in the transformer-based BERT network, using matrix arithmetic and trivial ReLU operations. Experiments on CPUs and GPUs show that this substitution significantly improves computational efficiency.

It is important to note that the aforementioned research, including our work, is orthogonal to efforts aimed at reducing DNN computation, such as network pruning (Mitchell A.Gordon and Andrews, 2020), compression (Thierry Tambe et al., 2021), and early exit (Li et al., 2022). Our method can be applied in conjunction with these methodologies to obtain further computational benefits derived from efficient GEMM computation platforms.

### 2.2 GENERAL MATRIX MULTIPLICATION ACCELERATOR

General Matrix Multiplication (GEMM) accelerators are specialized hardware components designed to expedite matrix multiplication operations(Kwon et al., 2019). They are employed in data centers for high-performance computing and edge devices to enhance efficiency in tasks such as digital signal processing, artificial intelligence, and scientific simulations(Qin et al., 2020). GEMM accelerators

can be integrated with devices like Tensor Processing Units (TPUs)(Jouppi et al., 2017), included in System-on-Chip (SoC) configurations(Mitra et al., 2014), or developed as standalone chips (Reggiani et al., 2023).

In a nutshell, GEMM comprises numerous parallel processing elements (PEs) and hierarchical memory (i.e., L1, L2, and L3), as shown in Fig. 1. Within each PE, the multiply-and-accumulate (MAC) unit performs computations, while the local (L1) buffer stores the MAC unit's inputs and partial sums. Multiple PEs are arranged in an array to enable parallel execution of many MAC operations. The network-on-chip (NoC) facilitates data transmission between the local (L1) buffers inside each PE and the global (L2) buffer for GEMM. Additionally, a high-bandwidth off-chip (L3) memory serves as a backup for providing input data and holding output data. Because data access energy and latency increase linearly from L1 to L2 and become orders of magnitude larger in L3 (Kwon et al., 2019), GEMM accelerators are usually designed to maximize data reuse within on-chip L1 and L2 buffers.

Compared to the general-purpose processor CPUs and GPUs, which accommodate a variety of instructions through a range of logic and arithmetic components, GEMM accelerators are explicitly designed for matrix multiplication using only MAC units and buffers. This focused approach to matrix multiplication results in exceptional efficiency (Hojabr et al., 2021). However, the GEMM accelerator can only process the general matrix multiplication computation. Additional special function units have to be located alongside the GEMM accelerator to process the other types of computations (Jouppi et al., 2017; Mitra et al., 2014). The types and numbers of special function units are carefully tailored to the computations of the targeted neural network models Pati et al. (2021).

In this paper, we propose NeuralMatrix to compute the versatile computations all with linear matrix operations. The NeuralMatrix enables running versatile neural networks on one GEMM accelerator by eliminating the limitations of deploying the special function units for various computation types. This approach endows different neural network models with application-specific efficiency, which is conventionally only available with application-specific accelerators.

## 3    NEURALMATRIX – COMPUTING NETWORKS WITH MATRIX OPERATIONS

This section describes how NeuralMatrix runs a series of decision and computation processes to map and compute the neural networks with linear matrix operations. Its high-level logic is depicted by the flow-chart in Fig. 2. First, the computation in neural networks can be classified into linear and nonlinear operations. Linear operations are directly mapped to GEMM accelerators through GEMM mapping (§ 3.1). Among nonlinear operations, NeuralMatrix will then decide if one operation already corresponds to a piecewise linear function (e.g., ReLU), which can be computed using the piecewise linear calculation method. If not, an offline piecewise approximation will be performed before it can be handled by piecewise linear calculations (§ 3.2). To support the network inference accuracy after the aforementioned approximation and finetuning, we introduce two training approaches for NeuralMatrix, specifically the *post-approximation* and *pre-approximation* methods, and discuss their potential impact on the final inference accuracy (§ 4).

### 3.1    MAPPING LINEAR OPERATIONS TO GENERAL MATRIX MULTIPLICATION

Linear operations are pervasive in DNNs, for example in fully connected layers, convolution kernels, attention mechanisms, and more. These linear operations involve 2D, 3D, or higher-dimensional tensors. We first introduce our GEMM mapping module, which is the foundation of NeuralMatrix since it is the interface to the GEMM accelerators. By applying reshaping and re-blocking techniques, these linear operations can be represented as matrix addition and multiplication operations with various sizes. For instance, in convolutional neural network (CNN) models, the convolution and fully connected layers are the main linear operations that can be transformed into matrix multiplication by reshaping the input features and filters into two matrices. The dimensions of these matrices are determined by the width, height, and number of channels in the original convolution computation.

Given that each GEMM accelerator has its own computational and memory capabilities, matrices of different sizes—reshaped from linear operations in DNNs—are processed block-wise on the GEMM accelerator. In other words, the input and weight matrices are partitioned into smaller blocks to compute the output matrix, taking advantage of the GEMM accelerator's three-level memory

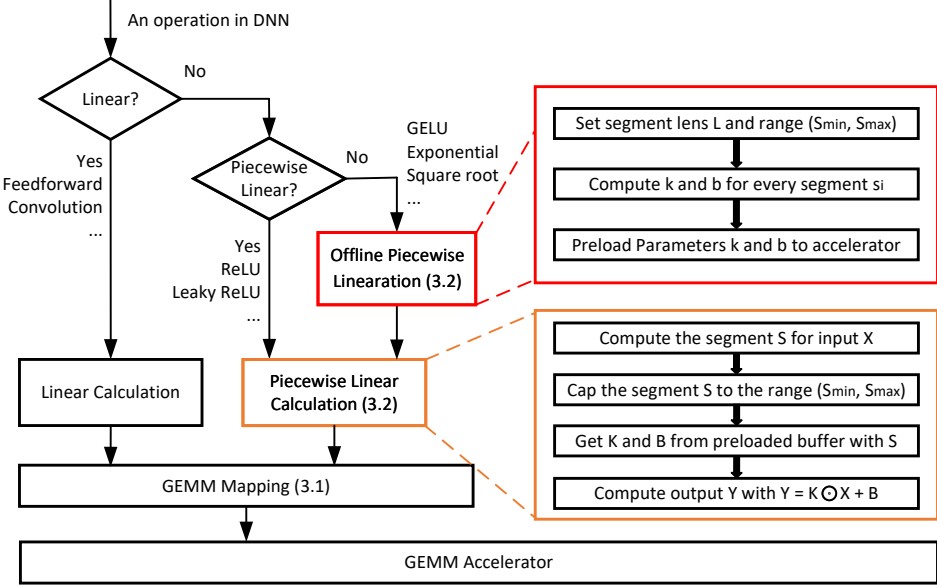

Figure 2: Overview of NeuralMatrix. Different types of DNN computation will go through different decision and process steps. Eventually, an entire neural network can be moved to linear matrix operations and become fully supported by a GEMM accelerator.

hierarchy to minimize energy consumption and buffer access times (Kwon et al., 2019). The optimal block division is achieved by exploring data flows using a top-down approach: first addressing the stationary scheme, followed by spatial/temporal accesses, and finally determining the tile size to find the optimized data flow of linear operations. The term "stationary" refers to storing matrix data in global and local buffers for extended periods to maximize its reuse. Data reuse can be classified into temporal and spatial reuse. Temporal reuse involves reading data from off-chip DRAM in chronological order, sending it to multiple local buffers, and performing multiplication or addition operations on the partial sums in processing elements (PEs). Conversely, spatial reuse entails moving and processing data in parallel. Lastly, the tile size defines the data size for each movement and computation.

The above division uses a method similar to grid search to find this optimal block division. For example, given a matrix multiplication with dimension $(M \times K) \times (K \times N)$, we change the block size in the three dimensions (stationary, spatial/temporal accesses, and tile sizes) from 2 to 128 with stride 2, and use an early stage model to calculate the latency and energy consumption of GEMM accelerator. Then we will choose the optimal block size in three dimensions with the minimum latency under energy consumption constraints.

### 3.2 MOVING NONLINEAR OPERATIONS TO MATRIX OPERATIONS

Addressing the nonlinear operations inherent in DNNs poses a significant challenge, as they cannot be easily mapped to standard matrix multiplications performed using GEMM accelerators, as described in the previous subsection. To overcome this issue, ASICs are specifically designed. These designs typically involve creating a predetermined number of functional units of various types, meticulously tailored to fulfill the nonlinear computational requirements of a specific neural network model. While this approach yields highly efficient accelerators adept at handling particular networks, it falls short in terms of flexibility. In contrast, our proposed method emphasizes enhanced adaptability, allowing a wide range of DNNs to be efficiently executed on a single GEMM accelerator.

**Offline Piecewise Linearization.** In NeuralMatrix, continuous nonlinear operations are approximated using piecewise functions. This method involves dividing a nonlinear function of interest into smaller regions within a chosen interval and approximating each region with a simpler function, such as a line or polynomial. Continuous piecewise linear (CPWL) approximations specifically utilize

Table 1: Parameter overhead in NeuralMatrix (granularity=0.25).

| DNN Model | ResNet-50 | BERT-base | GCN |
|---|---|---|---|
| Extra Parameter Size (FP16) | 114KB | 49.2KB | 0.24KB |
| Extra Parameter Size (INT8) | 57KB | 24.6KB | 0.12KB |
| Normalized Parameter Size | 0.46% | 0.01% | 0.10-0.74% |

lines for approximation, ensuring that the endpoint of one region is the same as the starting point of the next region.

There are two primary advantages of employing CPWL approximation in NeuralMatrix. Firstly, classic GEMM accelerators can support the computation in CPWL without any hardware modifications, unlike some other approximation methods, such as look-up table (LUT) based nonlinear approximation, which require extra hardware resources. Secondly, alternative approximation methods like Taylor expansion or Chebyshev approximation necessitate considerable additional computations, which do not align with our ultimate goal of computing efficiency.

**Piecewise Linear Calculation.** Technically speaking, piecewise linear operations from the above approximation are still nonlinear and can not be directly mapped to GEMM. Therefore, we develop a three-step approach here that can handle piecewise linear calculations. Here we use piecewise linearized GELU as an illustrative example (in Fig. 2), but the same process can also be used to handle operations that are originally piecewise linear, such as ReLU and Leaky ReLU. The input data is $X$ and the goal is to calculate $Y = \text{GELU}_{approx.}(X)$, $X, Y \in \mathbb{R}^{M \times N}$. The pre-calculated parameter $k$ and $b$ of each segment are pre-stored in off-chip DRAM, and indexed by the segment numbers.

Piecewise linear calculation follows the following steps: ① We use a linear operator to calculate the segment matrix $S$ for the input matrix $X$. Each element in $S$, e.g., $S_{i,j}$ represents which segment its corresponding input value $X_{i,j}$ falls into. The calculation of the segment matrix $S$ is handled by the GEMM accelerator and the output is further passed to the L2 global buffer [1]; ② The parameters $k$ and $b$ are aggregated and sent to the GEMM accelerator in the forms of slope matrix $K$ and intercept matrix $B$; ③ Finally, the GEMM accelerator performs element-wise calculations $Y = X \cdot K + B$ to get the output matrix $Y$. The other continuous nonlinear functions, such as softmax and layer normalization, can be computed by approximating inverse proportional, root, and exponential functions.

**Parameter Overhead.** NeuralMatrix introduces additional overhead to the original model, resulting from the extra parameters used to represent piecewise linear functions (e.g., $k$, $b$ tables). In this section, we provide a quantitative analysis of this aspect. Specifically, we focus on the scenario where the granularity is set to 0.25, and the parameter overhead is presented in Table 1. Since we employ fixed granularity for the entire network, utilizing larger granularity values will proportionally decrease the parameter overhead. For instance, if the granularity is doubled to 0.5, the overhead will be reduced by half. Even with the smallest granularity in our experiments, the parameter overhead remains minimal (less than 0.7%). Therefore, we will utilize 0.25 as the default segment granularity in the following sections.

## 4    APPROXIMATION WITH FINE-TUNING

NeuralMatrix primarily focuses on improving the inference-time efficiency of fine-tuned DNNs on resource-limited platforms. One approach, which we referred to as the *post-finetuning* method, involves standard fine-tuning of the original DNN. Following this, the DNN is transferred to linear matrix operations with necessary approximated piecewise linearization, as detailed in the previous subsection.

In addition, NeuralMatrix can be seamlessly integrated with training, offering an alternative *pre-finetuning* approximation approach. This technique involves mapping a pre-trained DNN to its

---

[1] When this piecewise linear approximation is calculated, the $k$ and $b$ parameters for all the segments $S_{i,j}$ used in this round of approximation are prefetched from DRAM to the L2 global buffer.

Table 2: The parameters of the implemented GEMM accelerator.

| Parameters | Total PE Numbers | Cluster Numbers | Stationary Scheme | Frequency |
|---|---|---|---|---|
| Value | 1024 | 8 | Output stationary | 1 GHz |
| Parameters | L1 buffer Size | L2 buffer Size | NoC Bandwidth | DRAM Bandwidth |
| Value | 4 KB | 1 MB | 128 Gbps | 32 Gbps |

approximated form before finetuning it on specific downstream tasks. The loss function used during finetuning remains unchanged from conventional finetuning, and standard automatic differentiation techniques can be employed for back-propagation. Both the post- and pre-finetuning methods yield final approximated DNNs with identical architectures and, consequently, have the same inference time cost on GEMM.

Any approximation method that maps DNNs, including nonlinear functions, to the GEMM accelerator will inevitably lead to a loss in computational accuracy, which in turn results in end-to-end inference accuracy loss. In the piecewise approximation of nonlinear operations, the accuracy of the approximation is dependent on the granularity of linear segments. Finer granularity contributes to higher accuracy but increases the number of parameters stored in the DRAM. In the following Section 5.2, we will demonstrate how to find a suitable tradeoff and select the appropriate linear segment granularity to achieve low memory cost and high inference accuracy for various downstream tasks.

## 5 EVALUATION

In this section, We first verify the inference accuracy after differnet DNNs are transformed to the linear matrix operations by the proposed NeuralMatrix. Next, we compare the computation efficiency of NeuralMatrix on a FPGA implemented GEMM accelerator with existing general-purpose and application-specific computation platforms to showcase the computation efficiency of NeuralMatrix.

### 5.1 EXPERIMENTAL SETUP

In this study, we first implement a systolic array based GEMM accelerator with the Xilinx Virtex 7 XC7VX485T FPGA. A data addressing unit in the global buffer (L2) is added to access the parameters $k$ and $b$ according to the segment number $S_{i,j}$. The architectural-level parameters, including the number of processing elements (PEs) and memory bandwidths of the GEMM accelerator, are optimized with the GEMM design automation tool (Wei et al., 2017). The parameters are summarized in Table 2. We choose the output stationary data movement strategy as it avoids frequent data movements to and from memories and benefits the lowest energy consumption for large matrix multiplication (Zhao et al., 2022). To assess the computation performance of CPUs and GPUs, we conducted tests on the Intel i7-11700 CPU, NVIDIA 3090Ti GPU, and Jetson Orin SoC, utilizing an IT9121 power analyzer. For the existing accelerator solutions, we gathered relevant data from published papers. To standardly and normally compare the computation performance across different network models and hardware processors. Our analysis mainly focuses on the computation efficiency, which is indicated by the computation throughput (i.e., integer or floating-point operations per second) with the power consumption from the hardware processor.

### 5.2 INFERENCE ACCURACY

Before demonstrating the advantage of computation efficiency in NeuralMatrix, we first empirically verify its inference accuracy through three popular DNN architecture categories and seventeen tasks of different natures. We ensure that applying NeuralMatrix with the appropriate setup only compromises ignorable accurate loss compared with the original models' final performance. Fig. 3 displays the final inference accuracy of various DNN architectures on multiple benchmark datasets. In this experiment, we select some of the most well-known pre-trained DNNs in each category, such as ResNet-50 (He et al., 2016), BERT-base (Devlin et al., 2018), and GCN (Kipf and Welling, 2016), to represent CNN, transformer, and GNN, respectively. Although we experimented with 17 benchmark datasets, due to space limitations, we only showcase four benchmarks for each DNN category and include the rest in

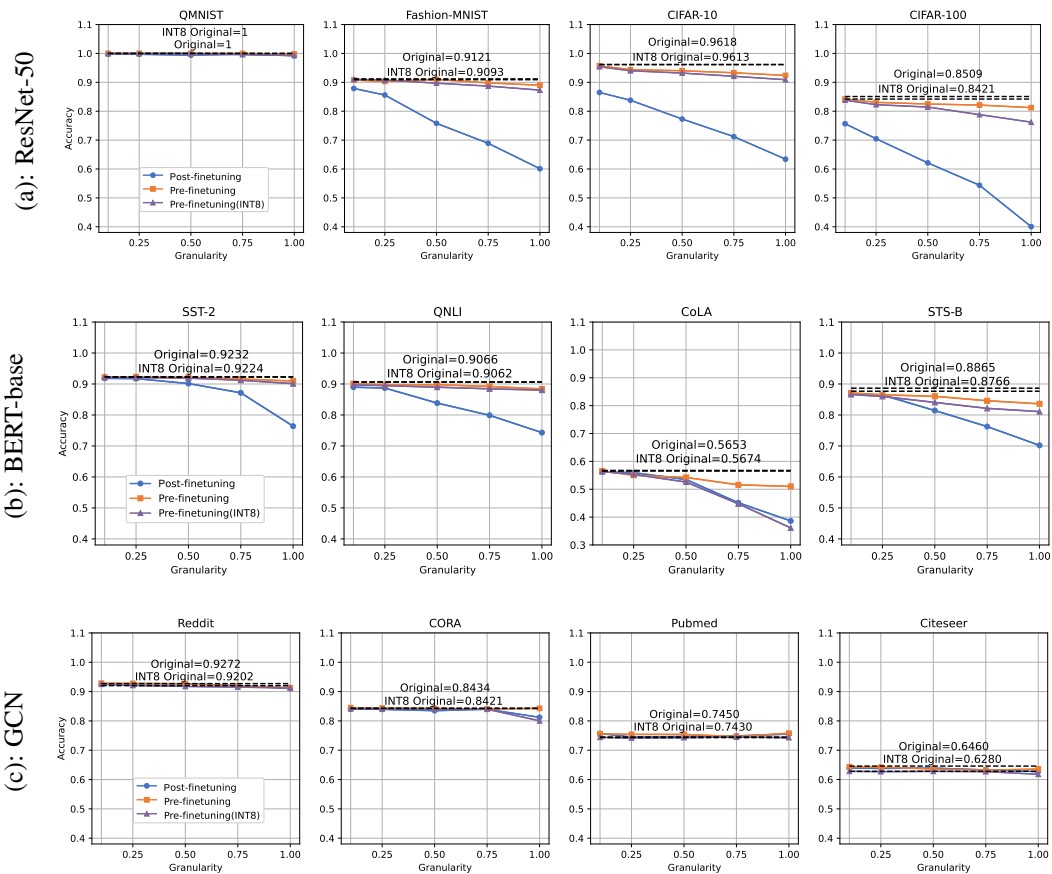

Figure 3: The network inference accuracy across benchmarks, granularity and training approaches

the following open-sourced NeuralMatrix. The dashed lines in each figure represent the inference accuracy of the original floating-point and quantization DNN models, which can be considered as the performance upper-bound given the pre-trained network. The sub-figures illustrate inference accuracy across different approximation granularity on different datasets. Specifically, the colored lines with markers represent the post-finetuning, the pre-finetuning performance using floating-point and INT8 quantization, respectively.

From the figures, we observe that the best-performing NeuralMatrix attains comparable performance to the original DNNs in every benchmark across different DNN categories. We find that pre-finetuning consistently achieves better performance than the post-finetuning approach, possibly because the network during finetuning is identical to the final inference ones. Thus, we recommend using the pre-finetuning method when transitioning DNNs to linear matrix operations. As the granularity increases from 0.1 to 1.0, we notice inference accuracy declines across all NeuralMatrix variants, which is expected since larger granularity incurs higher approximation errors. Generally, accuracy only begins to drop gradually when the granularity of exceeds 0.5 with floating-point and 0.25 with INT8 quantization. This demonstrates that a reasonably small granularity can achieve desirable performance. In the following evaluations, we use 0.25 as the default granularity.

Further, we experimentally test and compare the inference accuracy of NeuralMatrix (at the granularity of 0.25) across different network sizes, including depth and width, as well as variant network models. The inference accuracy of NeuralMatrix is summarized in Table 3 for different sizes of ResNet, BERT, GCN, respectively. Among all the tests, the NeuralMatrix introduces an average inference accuracy loss of 1.32%.

From the accuracy results, it is evident that the accuracy of NeuralMatrix is relatively robust to various network sizes and variant models of classic networks.

Table 3: Inference accuracy (%) of NeuralMatrix compared with original DNN models of different categories and sizes. We use CIFAR-10, SST-2 and CORA datasets for CNN, Transformer and GNN respectively.

| DNN Category/Dataset | Models | Size | FP16 | | INT8 | |
|---|---|---|---|---|---|---|
| | | | Original | NeuralMatrix | Original | NeuralMatrix |
| CNN/CIFAR-10 | ResNet-18 | 44.8 MB | 95.02 | 92.87 (-2.13) | 94.66 | 92.12 (-2.54) |
| | ResNet-34 | 85.3 MB | 96.12 | 94.85 (-1.27) | 96.10 | 94.43 (-1.67) |
| | ResNet-50 | 90.1 MB | 96.18 | 95.67 (-1.77) | 96.13 | 94.43 (-2.05) |
| | ResNet-101 | 163 MB | 97.13 | 95.67 (-1.51) | 96.44 | 94.82 (-1.62) |
| | ResNet-152 | 223 MB | 97.29 | 95.12 (-2.17) | 96.87 | 94.38 (-2.49) |
| Transformers/SST-2 | ALBERT | 47.4 MB | 92.50 | 88.42 (-4.13) | 89.76 | 85.28 (-4.48) |
| | DistilBERT | 263 MB | 90.41 | 88.65 (-1.35) | 88.19 | 86.24 (-1.95) |
| | BERT-Base | 436 MB | 92.32 | 92.32 (-0.00) | 92.24 | 92.07 (-0.25) |
| | BERT-Large | 1340 MB | 93.46 | 93.02 (-0.34) | 93.12 | 92.20 (-0.92) |
| GNN/CORA | GCN (L=1) | 94.0KB | 72.90 | 72.34 (-0.56) | 72.37 | 71.84 (-0.53) |
| | GCN (L=2) | 95.6KB | 84.28 | 84.31 (+0.03) | 83.95 | 84.02 (+0.07) |
| | GCN (L=3) | 99.4KB | 84.34 | 84.38 (+0.04) | 84.21 | 84.11 (-0.10) |
| | GCN (L=6) | 113.9KB | 81.18 | 81.11 (-0.07) | 80.86 | 80.92 (+0.06) |
| | GCN (L=9) | 128.3KB | 80.53 | 80.58 (+0.05) | 79.82 | 79.91 (+0.09) |

Regarding specific DNN models, we observe that for both BERT and ResNet, the performance gap between different NeuralMatrix variants increases as the baseline performance decreases. This suggests that one can choose a larger granularity for easier tasks but a smaller one for more difficult tasks. In contrast, the GCN models do not exhibit significant differences among the baseline and various NeuralMatrix variants, possibly because GCNs are typically shallower.

## 5.3 BENEFITS OF COMPUTATION EFFICIENCY

This section present the potential computation and power efficiency gains achievable through the migration of neural networks to linear matrix operations executed on a General Matrix Multiplication (GEMM) accelerator. We undertake a comparative analysis of the computation efficiency involved in running neural networks on a variety of processing units, including general-purpose CPUs, GPUs, System-on-Chips (SoCs), application-specific FPGA-based ASIC designs, and the NeuralMatrix with an FPGA-implemented GEMM accelerator. For Graph Neural Networks (GNNs), we restrict our evaluation to the above inference accuracy, as we encountered limitations in locating standardized ASIC designs for GNNs due to their numerous variants. To ensure a fair comparison of computation efficiency across diverse network models on different hardware processors, we meticulously document both the computation throughput (i.e., integer or floating-point operations per second) and the power consumption. A higher throughput with a smaller power consumption indicates a more efficient computing approach.

Figure 4 illustrates the computation efficiency (recorded with 1/throughput and power) of different networks (CNN, BERT, and GCN) and their variants on different hardware processors. Each point indicates a network variant on a hardware processor. The processor types are distinguished by the marker shapes. All the design points are scatter-plotted, and the Pareto frontiers consist of the optimal design points. We distinguish the designs by the hardware processor types. Clearly, across all the network models, the general-purpose processors CPU and GPU, especially the CPU, are located far away from the Pareto frontiers, indicating a low computation efficiency. This is because the general-purpose processors trade the computation efficiency for generality, which needs the support of additional software like software and hardware overheads. To quantify the improvement with CPU GPUs, SoC, the NeuralMatrix improves computation efficiency (i.e., throughput per power) by 49.58, 8.50, and 2.35 times.

The plots also indicate the related FPGA-based ASIC designs for these networks (ResNet and BERT) (Lian et al., 2019; Shen et al., 2017; Bai et al., 2018; Pham et al., 2012; Khabbazan and Mirzakuchaki, 2019; Su et al., 2018; Jang et al., 2019; Zhao et al., 2023; Li et al., 2021; Qi et al., 2021; Khan et al., 2021; Wang et al., 2021; Zhang et al., 2021; Ham et al., 2020). Compared to ASIC designs for networks of different sizes, NeuralMatrix can achieve the same level of computation efficiency distributed along the Pareto frontiers.

When we compare the computation efficiency of NeuralMatrix across different network variants, we find that the computation efficiency of NeuralMatrix increases as the networks become wider and

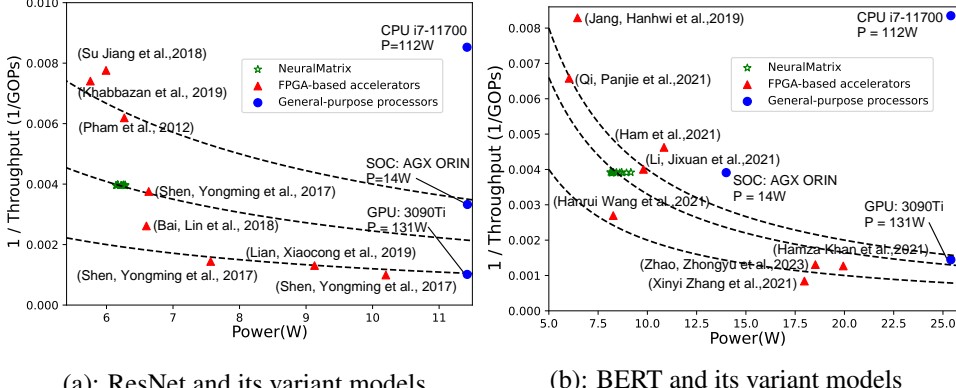

(a): ResNet and its variant models    (b): BERT and its variant models

Figure 4: Different networks' computation efficiency (throughput and power consumption) on different computing processors.

deeper. Network width has a greater impact on computation efficiency compared to depth. Later, the computation efficiency reaches the peak and stable range for the network with large sizes. According to this design space exploration, the NeuralMatrix will be more effective for the large neural network, especially the wider networks. This is because a wider neural network will be generated to a larger matrix, allowing more parallel computation to be performed simultaneously by the parallel processing elements in the GEMM accelerator. We can conclude that by transforming the computation of the entire neural network to general matrix multiplication, NeuralMatrix enables versatile neural networks to be efficiently executed by one GEMM accelerator. Meanwhile, NeuralMatrix demonstrates its superior efficiency for large neural networks, which can fully utilize its PEs after being transformed to matrix multiplication by NeuralMatrix.

## 6   DISCUSSION AND LIMITATIONS

In this study, we concentrated on three widely-used DNN backbone models and employed fixed approximation granularities for the entire DNN as a demonstrative example to highlight the benefits of transitioning entire neural networks to linear matrix operations. Nevertheless, we believe that NeuralMatrix can be applied to broader categories of network architectures as well. Moreover, by examining different nonlinear functions and their positions, it may be feasible to further minimize the accuracy loss of NeuralMatrix by employing varying approximation granularities (Hamann and Chen, 1994). Due to time and space constraints, we only implemented and tested the piecewise linear approximation method with the fixed granularity for the entire neural work; however, different granularities for different computation types or alternative approximation methods might potentially enhance network inference accuracy at the expense of increased computation cost. Furthermore, our evaluation of scalability across network sizes revealed that larger (deeper or wider) networks demonstrate greater computation efficiency gains. Moving forward, we also envision the proposed NeuralMatrix being effectively utilized in the inference process and the training process of large models in the future.

## 7   CONCLUSION

To overcome this limitation of numerous types of computation and enable versatile DNNs on a single computing units, we introduce NeuralMatrix. This approach transitions entire DNNs to linear matrix operations, which can further be be executed by general matrix multiplication accelerators. NeuralMatrix utilizes three mainstream DNN backbone models - CNN, transformer, and GNN - along with their variant models as illustrative examples. Our pre-finetuning training reveals that the shift to linear matrix operations incurs negligible inference accuracy loss. The evaluation demonstrates that NeuralMatrix can attain ASIC-level computation and energy efficiency on general-purpose matrix multiplication accelerators. Consequently, this enables the efficient support of a broad range of DNNs on a single hardware accelerator.

## 8 ETHICS STATEMENT

Our research aims to revolutionize the computation of complete Deep Neural Network (DNN) models with linear matrix operations and further execute on a General Matrix Multiplication (GEMM) accelerator. The significance of our work can be distilled into two key aspects. First, it achieves generality by enabling the execution of a diverse range of DNN models on a single GEMM accelerator, eliminating the necessity for additional or specialized functional units. Second, it maintains computational efficiency at a level comparable to what is conventionally achievable through accelerator designs tailored specifically to a single network model. In essence, our contribution facilitates the general-purpose ability to run versatile DNN models with the computational efficiency previously achievable to application-specific designs.

## 9 REPRODUCIBILITY STATEMENT

Our system design and implementation is reproducible. We have documented all the critical experimental details in the main text. While we cannot include the complete text of every design configuration and parameter due to their excessive length, We will open-source all of our implementation (including the code for computing and training the NeuralMatrix and also the RTL design of the implemented GEMM accelerator used in our evaluation etc.).

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
