# OpenReview forum: "NeuralMatrix: Compute the Entire Neural Networks with Linear Matrix Operations for Efficient Inference"
_ICLR.cc/2024/Conference — Submitted to ICLR 2024_

### Official Review · Reviewer_Zkh6 · 2023-10-27

**Soundness:** 3 good
**Presentation:** 3 good
**Contribution:** 3 good
**Rating:** 5
**Confidence:** 4

**Summary:**

The paper summarizes the approach of transforming non-linear operations into approximations through piecewise linear equations. This innovative technique enables the utilization of a single GEMM accelerator to enhance the acceleration of all DNN operations. By doing so, it not only improves the computational efficiency but also enhances the generality of the DNN operations.

**Strengths:**

* The explanation of the approach is presented with clarity. Its potential for generalization across a wide range of popular DNN models would make these models more hardware-friendly for efficient deployment.

* The implementation on an FPGA is also promising and demonstrate the completeness of the work.

**Weaknesses:**

* The approach appears robust when applied to smaller datasets such as MNIST, CIFAR-10 and SST. However, its performance on more complex tasks, such as Imagenet, wikitext and LRA, remains uncertain.
* Furthermore, the presentation of figures and tables is lacking in clarity, which may hinder the reader's understanding of the results.
* Can you add the implementation details of the accelerator on FPGA to the appendix? For example, logic utilization, clock frequency, etc.

**Questions:**

* Are Table 3 results based on the pre-fine tuning approach and 0.25 granularity? It is better to state that in the caption.
* Why are there three Pareto optimal front in Figure 4a and 4b? The data points of GPU are not on the right places.

**Details Of Ethics Concerns:**

No concerns.

---

### Official Review · Reviewer_JHrZ · 2023-10-27

**Soundness:** 2 fair
**Presentation:** 2 fair
**Contribution:** 2 fair
**Rating:** 5
**Confidence:** 4

**Summary:**

This paper proposes NeuralMatrix to transform the computations related to a DNN into linear matrix operations, and execute these linear matrix operations on a general-purpose matrix multiplication (GEMM) acclerator. The results demonstrate NerualMatrix can greatly improve the DNN inference throughput with only small accuarcy loss. NeuralMatrix approximates nonlinear operations by piecewise linear operations, and uses a GEMM accelerator to run the NeuralMatrix-approximated DNN inferences.

**Strengths:**

1. The paper explains the working mechanism of NeturalMatrix.
2. The paper flows well.

**Weaknesses:**

1. The paper does NOT compare NeuralMatrix against model compression, and quantization. Does directly pruning the nonlinear operations in a DNN or quantizing the nonlinear operations in a DNN can outperform NeuralMatrix?

2. The throughput values of NeuralMatrix can be added into Figure 3.

3. In Figure 4, Bai, Lin 2017 and Hanrui Wang 2021 are better than NeuralMatrix.

**Questions:**

Please comment on the 1-3 in the weakness section.

---

### Official Review · Reviewer_tPnn · 2023-10-31

**Soundness:** 2 fair
**Presentation:** 3 good
**Contribution:** 2 fair
**Rating:** 3
**Confidence:** 4

**Summary:**

The paper introduces NeuralMatrix framework to transform the computation of entire DNNs into linear matrix operations, enabling their execution with one general-purpose matrix multiplication (GEMM) accelerator. The performance is validated on ResNet, BERT, and GCN.

**Strengths:**

The proposed NeuralMatrix enables the execution of a diverse range of DNN models on a single GEMM accelerator, eliminating the necessity for additional or specialized functional units.

**Weaknesses:**

1.	NeuralMatrix requires mapping different computation types in DNN computation to linear matrix operations so that a single GEMM accelerator can fully support them, which is not trivial.
2.	Does it necessary to transform the simple computation (such as ReLU, LeakReLU) into GEMM? The computation complexity and power consumption of GEMM are much higher than original computation.
3.	The accuracy degradation of the NeuralMatrix seems significant for CNNs and ALBERT models.
4.	The computing power and computational energy efficiency ratio of NeuralMatrix are not outstanding.
5.	The performance comparison is limited. More typical NN processors should be compared, such as TPU.

**Questions:**

Please refer to weaknesses.

---

### Official Review · Reviewer_P9wc · 2023-11-01

**Soundness:** 3 good
**Presentation:** 3 good
**Contribution:** 2 fair
**Rating:** 5
**Confidence:** 4

**Summary:**

The paper introduces a new framework, NeuralMatrix, which transforms the non-linear operations in deep learning models into matrix operations. In particular, non-linear operations are converted into linear ones using Offline Piecewise Linearization and Piecewise Linear Calculation. The Inference accuracy of this modified deep learning model is assessed on various benchmarks, including the CIFAR-10 and CORA datasets. Furthermore, the hardware complexity of these networks is evaluated across CPU, GPU, and embedded processors.

**Strengths:**

1- The concept of converting all deep learning operations into matrix operations is interesting, as it reduces the need and cost associated with designing specific hardware for each model.
2- The paper is well-written and well-organized.

**Weaknesses:**

1- The accuracy degradation is significant even on smaller benchmarks like CIFAR-10. This raises the question of what the drop in inference accuracy might be when benchmarking on datasets like ImageNet or with larger deep learning models, such as large language models. A discussion on this topic from the author would be valuable.

2- A comparison with the ASIC designs of deep learning accelerators, as presented in references [1, 2, 3], is recommended.

[1] Yazdanbakhsh, Amir, Berkin Akin, and Kiran K. Seshadri. "An evaluation of edge tpu accelerators for convolutional neural networks." (2021).
[2] Chen, Yu-Hsin, et al. "Eyeriss v2: A flexible accelerator for emerging deep neural networks on mobile devices." IEEE Journal on Emerging and Selected Topics in Circuits and Systems 9.2 (2019): 292-308.
[3] Reuther, Albert, et al. "AI and ML accelerator survey and trends." 2022 IEEE High Performance Extreme Computing Conference (HPEC). IEEE, 2022.

**Questions:**

How does the novelty of the new approach compare to the following studies?

[1] Kim, Sehoon, et al. "I-bert: Integer-only BERT quantization." International Conference on Machine Learning. PMLR, 2021.

[2] Ming, Neo Wei, et al. "MA-BERT: Towards Matrix Arithmetic-only BERT Inference by Eliminating Complex Non-Linear Functions." The Eleventh International Conference on Learning Representations, 2022.

---

> ### Comment · Reviewer_P9wc · 2023-11-23
> **Reviewer comment**
>
> The deadline is almost approaching, and there has been no response from the authors. Therefore, I will not change my score.

---

### Official Review · Reviewer_jE62 · 2023-11-06

**Soundness:** 2 fair
**Presentation:** 2 fair
**Contribution:** 2 fair
**Rating:** 3
**Confidence:** 3

**Summary:**

The paper proposes NeuralMatrix, which aims to transform the computation of entire Deep Neural Networks (DNNs) into linear matrix operations. The goal of this transformation is to execute most DNN operators in GEMM kernels thus imrpoving computational effficiency. The evaluation on main stream DNN models and they discuss the inference accuracy aross different benchmarks.

**Strengths:**

GEMM is most widely used and optimized kernels in current deep learning community and converting all computation into GEMM would not only simplify the development pipeline, but also improve the overall efficiency.

NeuralMatrix allows for the efficient support of a broad range of DNNs, including popular models in CNNs / Transformers and GNNs. It also achieves the levels of computing efficiency with previous hardware that is specially designed and optimized.

**Weaknesses:**

The first concern is that whether the topic is suitable for ICLR conference. Though ICLR does accept papers in framework and system, hardware design papers usually go to HPCA / ISCA / Micro. Could the authors clarify why this paper is a good fit for ICLR?

The design principle for NeuralMatrix is to converting all operators into linear OPs and obviously such a conversion is lossy. This requires the converstion and pipeline to be carefully designed and throughfully tested. However, current inference accuracy evaluation only include CIFAR10 on CNN, SST-2 on Transformers, and CORA on GNNs , which is not enough to convince users.

**Questions:**

How is the convolution operations transformed into GEMM while preseving the efficiency?

The greenmark in Figure 4 is too small. Enlarging it would improve the presentation.

What is the cost of performance of post-finetuning? Would this be expensive for large language models?

---

> ### Comment · Reviewer_jE62 · 2023-11-21
>
> The discussion ddl is approaching and there is still no response. I will stick with my current rating.

---

### Meta-Review · Area_Chair_F2QK · 2023-12-17

**Metareview:**

This paper proposes to replace all neural network operations with GEMMs. Translating the diversity of neural network operations to those that operation would be incredibly helpful, since this is probably the most optimized of the operations for deep learning accelerators. With that said, the paper focused on small-scale networks rather than the large-scale ones where these transformations would be most important. None of the reviewers were in favor of acceptance, and the authors did not participate in the discussion process. I suggest rejection.

**Justification For Why Not Higher Score:**

The results were not convincing in their current form, and the authors did not particpiate in the discussion process.

**Justification For Why Not Lower Score:**

N/A

---

### Decision · Program_Chairs · 2024-01-16

Reject